# Determinants of early neonatal outcomes after emergency cesarean delivery at Hawassa University comprehensive specialised hospital, Hawassa, Ethiopia

Solomon Elias[1], Zenebe Wolde[2], Temesgen Tantu[3]*, Muluken Gunta[4], Dereje Zewudu[5]

1 Department of Gynecology and Obstetrics, Final Year Resident in Obstetrics and Gynecology in Hawassa University College of Medicine and Health Sciences, Hawassa, Ethiopia, 2 Department of Obstetrics and Gynecology in Hawassa University College of Medicine and Health Sciences, Hawassa, Ethiopia, 3 Department of Obstetrics and Gynecology in Wolkite University College of Medicine and Health Sciences, Wolkite, Ethiopia, 4 Wolaita Zone Health Department Officer, Wolaita Sodo, Southern Nations, Ethiopia, 5 Department of Anesthesia in Wolkite University College of Medicine and Health Sciences, Wolkite, Ethiopia

☯ These authors contributed equally to this work.
‡ These authors also contributed equally to this work.
* tematantu405@gmail.com

**Data Availability Statement:** All relevant data are within the paper and its Supporting Information files.

## Abstract

### Background

Neonatal mortality after cesarean delivery is three folds higher than mortality after vaginal births. Post cesarean early neonatal outcomes are associated with preoperative and intraoperative fetomaternal factors which are preventable in the majority of cases.

### Objective

To identify determinants of early neonatal outcomes after emergency cesarean delivery at Hawassa University Comprehensive Specialized Hospital, Hawassa, Southern Ethiopia.

### Method

Institution based cross sectional study was conducted on 270 emergency cesarean deliveries. Data were collected by using a pretested questionnaire by trained data collectors. Descriptive analysis was used to see the nature of the characteristics of interests. Pearson chi-square-test was used to check presence of association between independent and outcome variables. Bivariate analysis was used to sort out variables at p values less than 0.05 for multivariate logistic regression. Significance level was obtained using odds ratio with 95% CI and p value < 0.05.

### Results

The prevalence of adverse early neonatal outcome after emergency cesarean delivery was 26.7%. Around 11% of newborns had low (<7) fifth minute Apgar score and more than one-

**Funding:** Hawassa University funded the data collection and data entry during the study process. Otherwise, it had no role in study design, data collection, analysis, decision to publish, or preparation of the manuscript.

**Competing interests:** The authors have declared that no competing interests exist.

**Abbreviations:** NICU, neonatal intensive care neonate; WHO, world health organization; PNA, perinatal asphyxia; MSAF, meconium-stained amniotic fluid; C/S, cesarean section; LSFOL, latent phase first stage of labor.

third (34.8%) of them admitted to neonatal intensive care unit for more than 24 hours. Fifteen (5.6%) newborns died within their first seven days of life. Neonates with a preoperative meconium-stained amniotic fluid and low birth weight (< 2500 grams) had greater odds of having adverse early neonatal outcome with (AOR = 6.37; 95% CI: 2.64, 15.34) and (AOR = 14.00; 95% CI: 3.64, 53.84) respectively.

## Conclusion

The prevalence of adverse early neonatal outcome is high in this study and meconium-stained amniotic fluid during labor as well as low birth weight were the leading predictors of adverse early neonatal outcome during emergency cesarean delivery.

## 1 Introduction

Cesarean delivery is defined as the birth of a fetus via a surgically created incision in the anterior uterine wall [1]. Classification of cesarean section is based on the degree of urgency, women-based and others. Traditionally it is classified as, Elective cesarean delivery, if the decision to perform the operation was made ahead of time and /or before onset of labor and all others are considered as emergency cesarean delivery [2,3]. World health organization (WHO) sets a cesarean delivery rate of 5–15% that is assumed to be a range which can decrease neonatal morbidity and mortality [4,5]. Cesarean section may not be a measure of perinatal outcome, rather it is considered as a measure of a specific health care process (mode of delivery) [1].

The neonatal period is a highly vulnerable time for an infant completing many of the physiologic adjustments required for survival in the extra uterine environment. More specifically, the first seven days are the most critical period for the survival of newborns [6,7]. Different studies showed these poor neonatal outcomes were more common in emergency than elective cesarean delivery and are affected by various maternal and fetal factors [8–10].

Globally, around 2.6 million newborns died in 2016, which means 7,000 neonatal deaths every day [11]. Sub-Saharan Africa countries accounted 38% of all newborn deaths and Ethiopia is among the five countries which contribute 50% of global neonatal deaths [11].

Intrapartum related events accounts 24% of neonatal deaths and it is possible to prevent 80% of these deaths by having accessible and quality health services, of which, access to cesarean delivery is one component [11–15]. In sub-Saharan Africa, 8.8% of all deliveries are through cesarean section [16,17] and neonatal mortality after cesarean delivery in sub-Saharan Africa is higher than the global average [18]. Different types of factors such as parity, distance between institutions before interventions, meconium-stained amniotic fluid, type of anesthesia, birth weight, indications of cesarean sections are associated with poor neonatal out come after cesarean section across the literature [16,19–26].

There are limited studies done on factors which determine neonatal outcome after emergency cesarean section. These factors, which will be identified, might be modifiable and therefore, the finding of this study may be used as an input by health care providers and government bodies on measures needed to improve quality of care during pregnancy, labor and delivery.

## 2. Materials and methods

### 2.1 study area, design, and populations

Institution based cross sectional study was conducted at Hawassa university comprehensive hospital in Hawassa city 320 kilo meters south east to capital of Ethiopia from 01, August 2018 to 30, October 2018. All neonates who were delivered by emergency cesarian delivery at

Hawassa University comprehensive specialized hospital during the study period and who were eligible were included.

## 2.2 Sample size determination and sampling technique

Single population proportion formula was used to determine sample size required to conduct this study from previous studies with the prevalence of adverse early neonatal outcome after emergency cesarean reported as 20% at 95% certainty and 5% margin of error. Using this 20% proportion of adverse early neonatal outcomes, the calculated sample size for the first objective was 245 [27]. By a convenience sampling method, all neonates delivered by emergency cesarean deliveries in the study period were taken until the estimated sample size was achieved.

## 2.3 Inclusion & exclusion criteria

**2.3.1 Inclusion criteria.** Neonates who were delivered by emergency cesarean delivery and still births at cesarean delivery, who had a positive fetal heart beat pre operatively were included in the study.

**2.3.2 Exclusion criteria.** Neonates, who were delivered by emergency cesarean delivery with one the followings like who had a gross lethal congenital anomaly that was diagnosed pre operatively or post operatively, multiple pregnancy, intrauterine fetal death diagnosed on admission, and whose mothers declined or were unable to give medical history due to any medical disease or obstetric complication.

## 2.4 Data collection instruments

The mother was informed about the purpose and usefulness of the study then verbal consent was taken if she is willing to participate before commencement of data collection. Two trained midwives who had previous experience on data collection collected the data. They were supervised by a trained supervisor in order not to have difficulties during collection. Extensive data was collected on each woman who was included in the study and her new born through interview and by abstraction of relevant data from medical records. Neonates, who were admitted to NICU, who were discharged from maternity ward and who stayed in the maternity ward for maternal indication, were followed for the first 7 days for possible development of the other outcome variables through the course. Discharged Neonates, who were healthy, were followed by data collectors assigned to specific neonates (midwives) until the seventh day of life with a phone interview with the mother about the neonate like whether the neonate was breastfeeding, sleeping well, alert and if there was any problem, the mother had been informed to bring it back. A well-structured checklist written in English enquiring maternal and neonatal condition were prepared. The data included maternal socio-demographic variables, previous obstetric history, antenatal care, obstetric and medical complication, indication of cesarean delivery and intraoperative events like type of anesthesia and incision to delivery time. Data regarding neonatal status included sex, weight, first and fifth minutes Apgar, need for NICU admission for more than 24 hours and presence of death (still birth or early neonatal death).

## 2.5 data processing and analysis

The collected data was checked, and entered to Epi Info version 7 then exported to SPSS version 20 for further data cleaning and analysis. Frequency distributions were obtained to check for data entry error. Descriptive statistics, tables, graphs, means and frequency distribution were used to present the information. The presence of association between independent and outcome variables was checked by Pearson chi-square-test. Additionally, each independent

variable was fitted separately into bivariate logistic analysis to evaluate for degree of association with outcome variable. Thus, further degree of association was assessed by multivariate logistic regression on variables with p values less than 0.05. Significance level was obtained at odds ratio with 95% CI and p value < 0.05.

## 2.6 Ethical approval

Ethical clearance was obtained from the Institutional Review Board (IRB) of the college of medicine and health sciences, Hawassa University, Ethiopia with ethical clearance letter no. RPGe/88/2018. The IRB has given ethical clearance for both oral and written informed consent.

**Operation definitions.** Early neonatal outcome: Condition of a neonate in the first seven days after emergency cesarean delivery, which can be favorable or adverse.

Adverse early neonatal outcomes Stillbirthirth at delivery, fifth minute Apgar score < 7, admission to NICU for more than 24 hours and death within seven days are considered adverse early neonatal outcomes.

Good/favorable early neonatal outcomes: APGAR sore of >7 at the fifth minute, absence of admission to NICU or admission to NICU for < 24 hours are considered as good/favorable early neonatal outcomes.

Previous one cesarean scar with x factor: Cesarean scar with any other factors like malpresentation, prolonged latent phase, protracted or arrest of cervical dilatation and etc.

## 3. Results

### 3.1 Socio-demographic characteristics

All 270 mothers responded to the questionnaire, yielding a response rate of 100%. The mean age of participants was 26.4 (SD±4.8) years and ranged from 18 to 43 years. The vast majorities (98.1%) of respondents were married. Nearly half (47.4%) of the mothers were housewives, 20% of mothers never attended school and only 38 (17.6%) study subjects had degrees and above education level (Table 1).

### 3.2 Obstetric characteristics of participants

The mean gravidity and parity of participants was 2.4 (SD±1.5) and 1.3 (SD±1.49) respectively. Majority (84.1%) of the pregnancies were term at the time of admission. About 9%, 2.2% and 1.1% study subjects had history of abortion, stillbirth and early neonatal death respectively. (Table 2).

Preoperatively, pre-eclampsia (39.8%), premature rupture of membranes (16.9%) and abruption placenta (12%) were the most commonly diagnosed obstetric complications among the study subjects (Fig 1).

### 3.3 Preoperative characteristics of participants

Of the total respondents, more than half (55.9%) of them were referred from other institutions; majority (60.3%) from government hospitals. For those referred from other institutions, the mean time taken for transport was 2.12 (SD±1.723) hours. More than three-fourth (78.5%) participants had labor before operation, of whom, 194 (91.5%) presented with spontaneous onset of labor with a mean duration of labor being 11.2(SD±7.78) hours (Table 3).

The most common indications for cesarean delivery in the participants were one uterine scar with x-factor (14.8%), fetal bradycardia (13.7%), fetal tachycardia (13%), and Malpresentation (10.4%). (Fig 2).

**Table 1. Socio-demographic characteristics of mothers in Hawassa University comprehensive specialized hospital, Southern Ethiopia, October 2018.**

| Variables (n = 270) | Frequency | Percentage |
|---|---|---|
| Maternal age | | |
| 20 | 33 | 12.2 |
| 21–34 | 217 | 80.4 |
| ≥35 | 20 | 7.4 |
| Religion | | |
| Orthodox | 91 | 33.7 |
| Muslim | 105 | 38.9 |
| Protestant | 73 | 27.0 |
| Others | 1 | 0.4 |
| ➢ Marital status | | |
| Married | 265 | 98.1 |
| Unmarried | 4 | 1.5 |
| Divorced | 1 | 0.4 |
| Ethnicity | | |
| Sidama | 81 | 30.0 |
| Oromo | 129 | 47.8 |
| Wolayta | 4 | 1.5 |
| Halaba | 5 | 1.9 |
| Amhara | 23 | 8.5 |
| Gurage | 9 | 3.3 |
| Others | 19 | 7.0 |
| Residence | | |
| Rural | 87 | 32.2 |
| Urban | 183 | 67.8 |
| Ever attended school | | |
| Yes | 216 | 80.0 |
| No | 54 | 20.0 |
| Level of education (n = 216) | | |
| Elementary level | 84 | 38.9 |
| High school level | 69 | 31.9 |
| Diploma level | 25 | 11.6 |
| Degree level and above | 38 | 17.6 |
| Occupation | | |
| Farmer | 45 | 16.7 |
| Merchant | 36 | 13.3 |
| Gov't employee | 56 | 20.7 |
| House wife | 128 | 47.4 |
| Others | 5 | 1.9 |
| Monthly family income (Birr) | | |
| < = 1000 | 76 | 28.1 |
| 1001–5000 | 141 | 52.2 |
| >5000 | 53 | 19.6 |

## 3.4 Early neonatal outcomes

About 56% of the mothers were operated during duty hours. Sixty percent of surgeries were done after 30 minutes of decision and majority (93%) of the surgeries were done under spinal

**Table 2. Obstetric profile of respondents in Hawassa University comprehensive specialized hospital, Southern Ethiopia, October 2018.**

| Variables (n = 270) | Frequency | Percentage |
| --- | --- | --- |
| Gravidity | | |
| 1–4 | 237 | 87.8 |
| ≥5 | 33 | 12.2 |
| Parity | | |
| ≤3 | 241 | 89.3 |
| >3 | 29 | 10.7 |
| Gestational age | | |
| Preterm (< 37 weeks) | 21 | 7.8 |
| Term (37–42weeks) | 227 | 84.1 |
| Post term (> 42 weeks) | 22 | 8.1 |
| Previous history of abortion | | |
| Yes | 24 | 8.9 |
| No | 246 | 91.1 |
| Previous history of stillbirth | | |
| Yes | 6 | 2.2 |
| No | 264 | 97.8 |
| Previous history of early neonatal death | | |
| Yes | 3 | 1.1 |
| No | 267 | 98.9 |
| ANC in current pregnancy | | |
| Yes | 266 | 98.5 |
| No | 4 | 1.5 |
| Place of ANC visit (n = 266) | | |
| Health center or private clinic | 211 | 79.3 |
| Primary hospital | 46 | 17.3 |
| Referral Hospital | 9 | 3.4 |
| Previous uterine scar | | |
| Yes | 71 | 26.3 |
| No | 199 | 73.7 |
| Number of uterine scar (n = 71) | | |
| One | 51 | 71.8 |
| Two | 18 | 25.4 |
| Three and above | 2 | 2.8 |
| Preoperative obstetric complications | | |
| Yes | 83 | 30.7 |
| No | 187 | 69.3 |
| Preoperative medical complications | | |
| Yes | 9 | 3.3 |
| No | 261 | 96.7 |
| Type of medical complications (n = 9) | | |
| Diabetes mellitus | 2 | 22.2 |
| Chronic hypertension | 1 | 11.1 |
| Cardiac disease | 1 | 11.1 |
| HIV infection | 3 | 33.3 |
| Others | 2 | 22.2 |

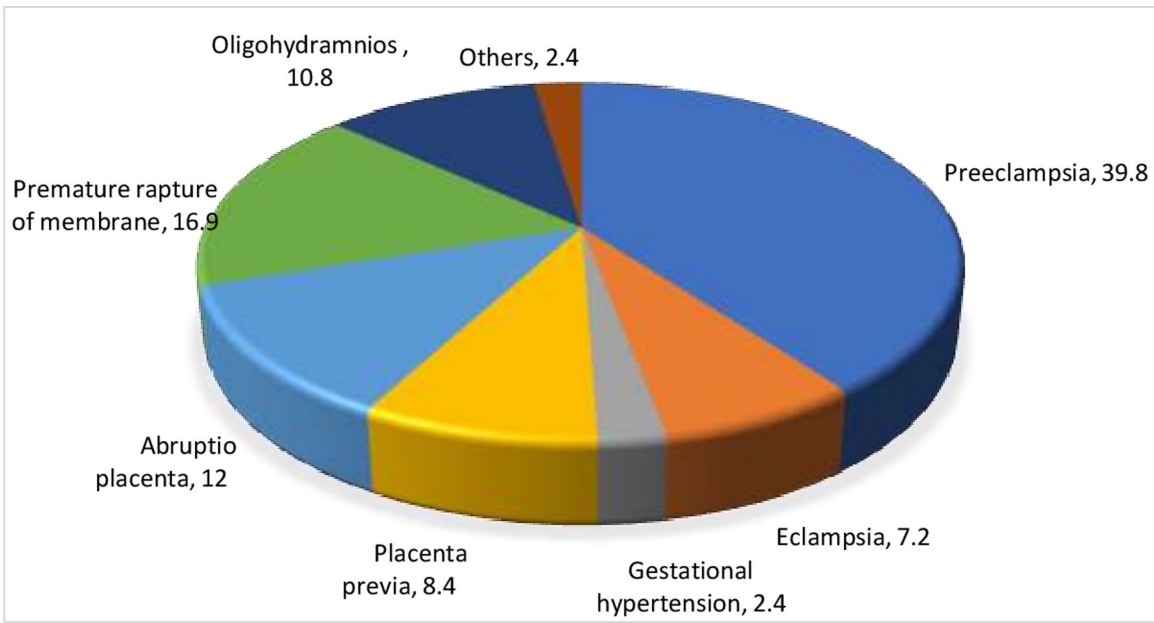

**Fig 1. Freqency of preoperative obstetric complications among mothers who had emergency cesarean delivery in HUCSH, 2018.**

anesthesia. Skin incision for most (94.1%) participants was supra pubic transverse type and year two and three residents operated on majority of the mothers (51.5% and 41.9% respectively). (See Table 4)

The prevalence of adverse early neonatal outcomes was 26.7% (95% CI: 21.3%-31.9%). Nearly 11% of the newborns had fifth minute APGAR score less than 7 and more than one-third (34.8%) were admitted to the NICU for more than 24 hours. Fifteen (5.6%) newborns died within their first seven days of life, of these, two-third of them died in the first postoperative day. The most common cause for early neonatal death was perinatal asphyxia (53.3%) followed by respiratory distress syndrome (40%).

### 3.5 Factors associated with early neonatal outcomes

In the binary logistic regression analysis, the following variables were statistically significant like: gestational age, previous cesarean scar, preoperative obstetric complications, admission type, place of referral, preoperative labor, stage of labor before operation, fetal heartbeat, type of anesthesia, income, parity, number of uterine scars, time spent from referring institution, liquor status and birth weight.

In the multi-variable logistic regression analysis, only liquor status and birth weight remained as statistically significant predictors of early neonatal outcomes. Mothers with meconium-stained amniotic fluid during vaginal examination were about 6-times (AOR: 6.34; 95% CI 2.64, 15.34) more likely to have adverse early neonatal outcomes than their counter-parts with clear amniotic fluid. Similarly, low birth weight increased the odds of adverse early neonatal outcomes by 14-fold (AOR: 14.00; CI 3.64, 53.84) (see Table 5).

### 4. Discussion

This study included 270 singleton neonates born via emergency cesarean delivery, of which 26.7% had adverse early neonatal outcomes (low fifth minute Apgar score, early neonatal death or admission to NICU for more than 24 hours). This finding was higher than Rwandan

**Table 3. Admission and preoperative characteristics of respondents in Hawassa University comprehensive specialized hospital, Southern Ethiopia, 2018.**

| Variables | Frequency | Percentage |
|---|---|---|
| Admission type (n = 270) | | |
| Not referred | 119 | 44.1 |
| Referred | 151 | 55.9 |
| Referring institution (n = 151) | | |
| Health center | 29 | 19.2 |
| Government hospital | 91 | 60.3 |
| Private clinic or Hospital | 31 | 20.5 |
| Travel time from referring institution | | |
| ≤1 hours | 195 | 72.2 |
| 2–3 hours | 56 | 20.7 |
| ≥4 hours | 19 | 7.0 |
| Labor before operation | | |
| Yes | 212 | 78.5 |
| No | 58 | 21.5 |
| Onset of labor (n = 212) | | |
| Spontaneous | 194 | 91.5 |
| Induced | 18 | 8.5 |
| Duration of labor before operation (n = 212) | | |
| < = 6 hours | 131 | 48.5 |
| 7–18 hours | 102 | 37.8 |
| >18 hours | 37 | 13.7 |
| Stage of labor at decision for CS (n = 212) | | |
| Latent first stage | 119 | 56.1 |
| Active first stage | 69 | 32.5 |
| Second stage | 24 | 11.3 |
| Liquor status at decision to operate (n = 212) | | |
| Clear | 118 | 55.7 |
| Grade one meconium-stained amniotic fluid | 12 | 5.7 |
| Grade two meconium-stained amniotic fluid | 19 | 9.0 |
| Grade three meconium-stained amniotic fluid | 35 | 16.5 |
| Unknown | 28 | 13.2 |
| Fetal presentation (n = 212) | | |
| Vertex | 183 | 86.3 |
| Brow | 2 | 0.9 |
| Face | 6 | 2.8 |
| Breech | 17 | 8.0 |
| Shoulder | 4 | 1.9 |
| Fetal heart beat at decision for operation (n = 270) | | |
| < 120 | 42 | 15.6 |
| 120–160 | 188 | 69.6 |
| > 160 | 40 | 14.8 |

and Nigerian study, which was 9% and 13.5% [19,28]. This variation could be due to a difference in the level of Hospital where studies were conducted. Rwandan study was done in a district Hospital, while this study was conducted in a tertiary Hospital, where more complicated pregnancies are referred to and handled. This is evidenced by higher proportions of referred

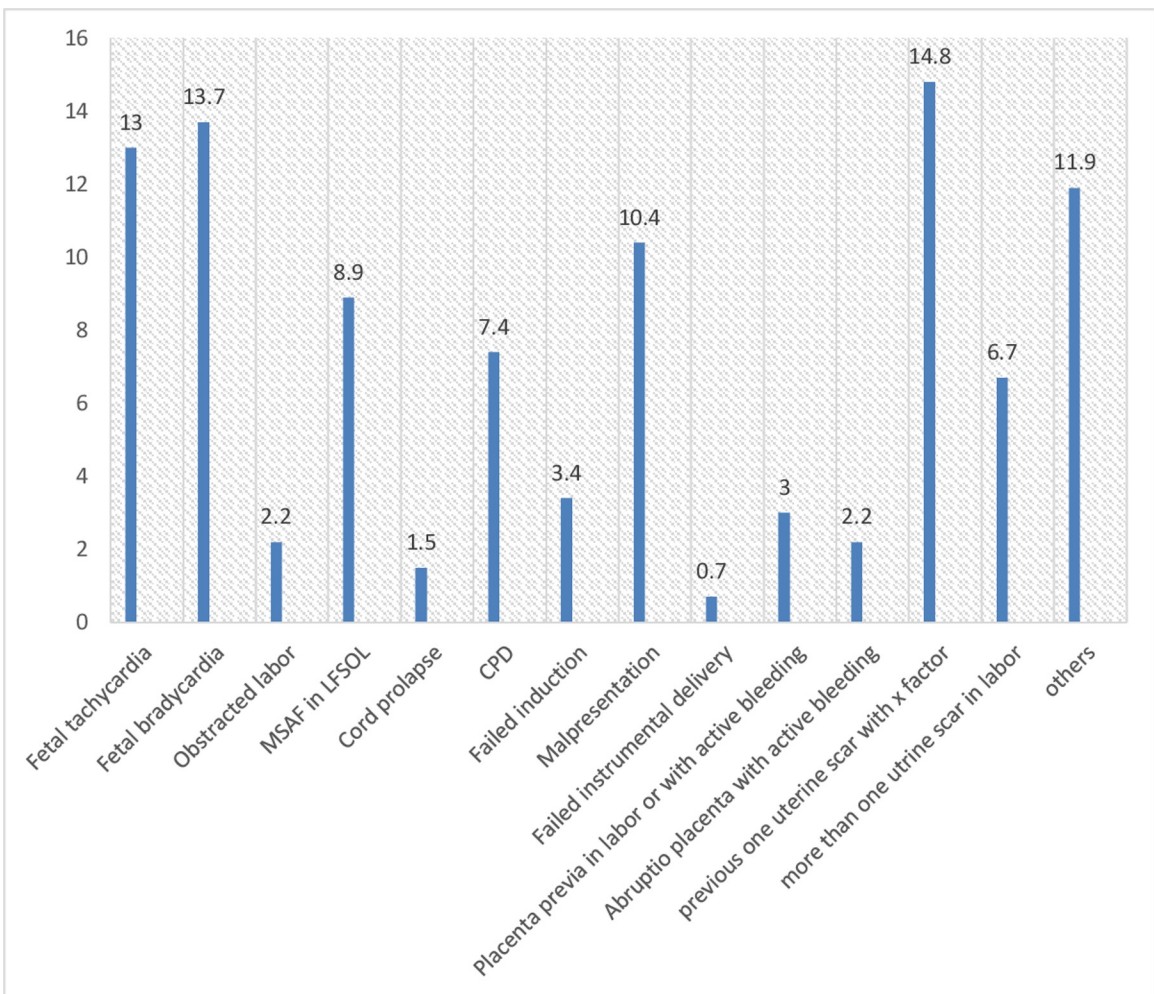

**Fig 2. Indications for cesarean delivery in women with emergency C/S in HUCSH, 2018.**

pregnant mothers, obstetric complications and severe forms of cesarean indications (tachycardia, bradycardia, cord prolapse and obstructed labor). Additionally, among referred mothers, proportion of patients with ambulance travel time greater than one hour was larger in this study. Additionally, majority of patients were in labor preoperatively and both duration of labor as well as ambulance travel time, which were associated with early neonatal outcomes, were larger in this study than the previous. This is also supported by a large difference in the proportion of meconium-stained amniotic fluid, a manifestation of utero-placental insufficiency, for which prolonged labor is among the known risk factors.

In the present study low fifth minute APGAR score (<7) was seen in 10.7% of deliveries which is in line with two African studies done in Nigeria (13.3%) and Rwanda (9%) [19,28].

About 34.8% of neonates were admitted to NICU for more than twenty-four hours and this is comparable with a magnitude reported in two Indian studies which were 32% and 26% [8,10]. In this study admission to NICU is higher than study done in Saint Paul's millennium medical college in Addis Ababa which can be explained by that Addis Ababa is Ethiopia's capital so referrals were from better setup and short distance [29]. The prevalence of admission to NICU in this study was higher than Australian study, which was 14.9% [9]. This might be the

**Table 4. Intraoperative characteristics and neonatal outcomes of participants in Hawassa University comprehensive specialized hospital, Southern Ethiopia, 2018.**

| Variables | Frequency | Percentage |
|---|---|---|
| Time of operation (n = 270) | | |
| Working hours | 119 | 44.1 |
| Duty hours | 151 | 55.9 |
| Decision to delivery time (n = 270) | | |
| ≤30 min | 108 | 40.0 |
| >30 min | 162 | 60.0 |
| Skin incision to delivery time (n = 270) | | |
| ≤5 min | 144 | 53.3 |
| >5 min | 126 | 46.7 |
| Type of anesthesia (n = 270) | | |
| Spinal anesthesia | 251 | 93.0 |
| General anesthesia | 19 | 7.0 |
| Maternal blood pressure at decision for operation (n = 270) | | |
| < 100/60 | 2 | 0.7 |
| 100–139/60–89 | 205 | 75.9 |
| 140/90 and above | 63 | 23.3 |
| Maternal blood pressure after anesthesia (n = 270) | | |
| < 100/60 | 9 | 3.3 |
| 100–139/60–89 | 226 | 83.7 |
| 140/90 and above | 35 | 13.0 |
| Type of skin incision (n = 270) | | |
| Suprapubic transverse | 254 | 94.1 |
| Midline | 16 | 5.9 |
| Surgeon (n = 270) | | |
| Year one resident | 7 | 2.6 |
| Year two resident | 139 | 51.5 |
| Year three resident | 113 | 41.9 |
| Year four resident | 10 | 3.7 |
| Obstetrician | 1 | .4 |
| Sex of neonate | | |
| Male | 148 | 54.8 |
| Female | 122 | 45.2 |
| Newborn weight | | |
| normal birth weight | 202 | 74.8 |
| low birth weight | 43(< 5th centile) | 15.9 |
| large birth weight | 25 | 9.3 |
| First minute Apgar score (n = 270) | | |
| < 7 | 45 | 16.7 |
| 7 and above | 225 | 83.3 |
| Neonate referred to NICU (n = 270) | | |
| Yes | 92 | 34.1 |
| No | 178 | 65.9 |
| Neonatal diagnosis at NICU (n = 92) | | |
| Preterm neonate | 29 | 31.5 |
| low birth weight | 11 | 12.0 |
| Meconium aspiration | 23 | 25.0 |

(*Continued*)

**Table 4.** (Continued)

| Variables | Frequency | Percentage |
|---|---|---|
| Perinatal asphyxia | 14 | 15.2 |
| Respiratory distress syndrome | 15 | 16.3 |
| Unfavorable early neonatal outcome | | |
| Yes | 72 | 26.7 |
| No | 198 | 73.3 |
| Fifth minute Apgar score (n = 270) | | |
| < 7 | 29 | 10.7 |
| 7 and above | 241 | 89.3 |
| Duration of admission at NICU (n = 92) | | |
| < 24 Hours | 32 | 34.8 |
| >24 hours | 60 | 65.2 |
| Neonatal condition on 7$^{th}$ postop day (n = 270) | | |
| Discharged improved | 203 | 75.2 |
| Alive and on treatment | 52 | 19.3 |
| Died | 15 | 5.6 |
| Cause of newborn death (n = 15) | | |
| Perinatal asphyxia | 8 | 53.3 |
| Respiratory distress syndrome | 6 | 40.0 |
| Meconium aspiration syndrome | 1 | 6.7 |
| Time of neonatal death (n = 15) | | |
| First post operation day | 10 | 66.7 |
| Second post operation day | 3 | 20.0 |
| Third post operation day | 2 | 13.3 |

fact that the Australian study was conducted in women with a better prepartum and Intrapartum care.

Regarding magnitude of early neonatal death (5%), this finding was comparable with a report from a study conducted at Attat Hospital, Ethiopia (3.6%) [30].

The current study showed that adverse early neonatal outcome was significantly associated with a preoperative meconium-stained amniotic fluid as compared with neonates delivered with a clear preoperative amniotic fluid. This finding is consistent with a prospective observational study conducted in India [20]. This could be explained by the increased risk of meconium aspiration syndrome and perinatal asphyxia in a meconium-stained amniotic fluid. In this study they contributed for 40.2% of NICU admission and 60% of early neonatal deaths respectively.

In this study, birth weight was associated with adverse early neonatal outcome. Neonates with low birth weight were 14 times more likely to have adverse early neonatal outcome when compared to those with normal birth weight. This is in line with a study conducted in Gondar university referral Hospital, where neonates with low birth weight had more adverse early neonatal outcome [26]. This could be explained by multiple factors. According to WHO global survey on maternal and perinatal health in seven African countries, tertiary or referral facilities having a significant amount of complicated pregnancies which necessitates premature termination of pregnancies, like preeclampsia, antepartum hemorrhage, preterm premature rupture of membranes, and preterm labor, will have a large amount of low birth weight delivery. As result, there will be increased neonatal complication rate as well as premature neonatal deliveries in those facilities [16]. Accordingly, this study was conducted in a tertiary hospital where

**Table 5. Binary and multivariable logistic regression analyses of factors affecting early neonatal outcomes in HUCSH, Southern Ethiopia, Oct 2018.**

| Variables | Early neonatal outcome n (%) | | COR (95% CI) | AOR (95% CI) |
|---|---|---|---|---|
| | Yes | No | | |
| Preoperative obstetric complications | | | | |
| Yes | 37 (51.4%) | 46 (23.2%) | 3.49 (1.98, 6.16) | 2.60 (0.92, 7.38) |
| No | 35 (48.6%) | 152(76.8%) | 1 | 1 |
| Admission type | | | | |
| Not referred | 22 (30.6%) | 97 (49.0%) | 1 | 1 |
| Referred | 50 (69.4%) | 101 (51.0%) | 2.18 (1.23, 3.87) | 0.96 (0.42, 2.20) |
| FHB at decision for operation | | | | |
| 120–160 beats/min | 43 (59.7%) | 145 (73.2%) | 1 | 1 |
| <120 beats/min | 19 (26.4%) | 23 (11.6%) | 2.78 (1.38, 5.59) | 2.29 (0.82, 6.39) |
| >160 beats/min | 10 (13.9%) | 30 (15.2%) | 1.12 (0.50, 2.48) | 1.52 (0.56, 4.16) |
| Type of anesthesia | | | | |
| Spinal | 62 (86.1%) | 189 (95.5%) | 1 | 1 |
| General | 10 (13.9%) | 9 (95.5%) | 3.38 (1.31, 8.71) | 1.69 (0.40, 7.01) |
| Number of uterine scar | | | | |
| One | 7 (9.7%) | 64 (32.3%) | 1 | 1 |
| Two or more | 65 (90.3%) | 134 (67.7%) | 4.43 (1.92, 10.21) | 2.90 (0.79, 10.66) |
| Liquor status | | | | |
| Clear | 21 (42.0%) | 125 (77.2%) | 1 | 1 |
| Meconium stained | 29 (58.0%) | 37 (22.8%) | 4.66 (2.38, 9.12) | **6.37 (2.64, 15.34)**\* |
| Birth weight | | | | |
| Normal | 37 (51.4%) | 165 (83.3%) | 1 | 1 |
| Low | 32 (44.4%) | 11 (5.6%) | 12.97 (5.99, 28.08) | **14.00 (3.64, 53.84)**\* |
| large | 3 (4.2%) | 22 (11.1%) | 0.60 (0.17, 2.13) | 0.57 (0.11, 2.94) |

AOR adjusted odds ratio; CI confidence interval; COR crude odds ration

\*statistically significant variables at p-value <0.05; Hosmer and Lemeshow goodness-of-fit 0.59.

more complicated pregnancies are managed and this might end up with large number of low-birth-weight neonate.

## Conclusion

The magnitude of adverse early neonatal outcome is high in this study. Low birth weight and meconium-stained amniotic fluid were significant independent determinants of adverse early neonatal outcome.

## Limitation of the study

Since it was a cross sectional study may not show time association between factors and the outcome variables.

## Supporting information

**S1 Appendix. 1 s1 questionnaire for determinannts of early neonatal outcomes after emergency cesarean delivery at Hawassa University comprhenssive specialised hospital, Hawassa, Ethiopia.**
(DOCX)

**S1 Data.**
(SAV)

## Acknowledgments

**Consent**

After getting ethical clearance from the institutional review board of Hawassa University and clarifying the parents of study participants about the aim of the study, informed consent was taken from a parent of the study participants after informing them the aim of the study. They have been told as they can withdraw from the study at any step if they feel so as well as confidentiality was granted.

## Author Contributions

**Conceptualization:** Solomon Elias, Zenebe Wolde, Temesgen Tantu.

**Data curation:** Solomon Elias, Zenebe Wolde, Temesgen Tantu.

**Formal analysis:** Zenebe Wolde.

**Investigation:** Solomon Elias, Zenebe Wolde.

**Methodology:** Solomon Elias, Temesgen Tantu.

**Software:** Temesgen Tantu, Muluken Gunta, Dereje Zewudu.

**Supervision:** Muluken Gunta, Dereje Zewudu.

**Writing – original draft:** Dereje Zewudu.

**Writing – review & editing:** Temesgen Tantu, Muluken Gunta.

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
