## [Decision Letter · Decision Letter 0]

4 Jan 2022

PONE-D-21-28401DETERMINANNTS OF EARLY NEONATAL OUTCOMES AFTER EMERGENCY CESSARIAN DELIVERY AT HAWASSA UNIVERSITY COMPRHENSSIVE SPECIALISED HOSPITAL, HAWASSA, ETHIOPIAPLOS ONE

Dear Dr. Temesgen Tantu

Thank you for submitting your manuscript to PLOS ONE. After careful consideration, we feel that it has merit but does not fully meet PLOS ONE’s publication criteria as it currently stands. Therefore, we invite you to submit a revised version of the manuscript that addresses the points raised during the review process.

We look forward to receiving your revised manuscript.

Kind regards,

Francesca Crovetto

Academic Editor

PLOS ONE

https://journals.plos.org/plosone/s/file?id=ba62/PLOSOne_formatting_sample_title_authors_affiliations.pdf”

4. Thank you for stating the following in the Funding Section of your manuscript:

“The study data collection was funded by College of Medicine and Health Sciences, Hawassa University, Ethiopia. The funding body has no role in the design of the study and collection, analysis, and interpretation of data and in writing the manuscript”

**Comments to the Author**

Review Comments to the Author

**Reviewer #1**: Very informative study

1.It it possible to find out how many babies were FGR, <3rd centile and 3rd to 10th centile (Intergrowth 21) in place of just mentioning low birth weight <2500 grams.?

Just to know, like how many babies could have been delivered earlier to prevent intrapartum complications . how many babies were diagnosed and electively induced?

2. What do you mean by ANC care? How may visit?? were all participants booked adequately?

3. Were there any neonates with congenital malformation?? adding to the neonatal morbidity

**Reviewer #2:** GENERAL: There are several grammatical and spelling errors as well as wrong use of tenses that need to be corrected. The referencing of statements is not entirely sequential. In the introduction, the authors jumped from number 16 to 18 and then in the discussion number 26 to 28.There is no number 27.

TITILE: The spelling of caesarean should be corrected.

ABSTRACT: NICU should be written in full. In the conclusion, the 1st sentence is not clear as to what the authors are comparing their prevalence of adverse early neonatal outcome with. Did they mean -the prevalence of adverse neonatal outcome was high--?

INTRODUCTION: lines 2-3 1st paragraph is not clear. Paragraph 3, 1st sentence should be referenced.

METHODS: KM should be written in full 1st before abbreviating. Under exclusion criteria, what do the authors mean by negative foetal heart beat? Under data collection, the 1st sentence should be recasted. The authors should explain clearly how the collected information from the mothers, the type instrument used and when and how this was done. Prior to obtaining information form the women, they should have been educated and informed about the study. When and where was this done? In line 8, what do the authors mean by contact to maternal? The authors should explain how follow up of the mothers was done by phone, who made the phone call to all the mothers? There should be a statement on ethical considerations in the methods section.

RESULTS: What was the total number of mothers enrolled into the study? Under obstetric complications, premature rupture of membrane should be premature rupture of membranes. Several spelling errors that need to be corrected. MSAF and LFSOL should be written in full 1st. In table 4, how did the authors determine the causes of neonatal death, were autopsies conducted on all the babies that died?

DISCUSSION: In the last paragraph before the discussion, lines 6-11 are not clear and should be recasted.

CONSENT: This section is not clear and should be re-written for clarity. Who specifically did the authors obtain consent form and what type of consent was obtained? When were the mothers educated and informed about the study prior to their accepting to participate in the study and obtaining consent? Ethical considerations may be used as a subheading instead.

REFERENCES: Numbers 1, 10 and 30 do not appear complete

---

## [Author Response · Author response to Decision Letter 0]

19 Jan 2022

Response for the first reviewer

1. I humbly accept all comments and thank you for your constructive comment

• The cut point in classifying gestational age was as follows < 37 weeks, 37-42 weeks, and > 42 weeks as a result weight less than 2500 grams is considered as less than 5th centile of the gestational age estimation (37 weeks). That is why we used the term low birth weight for all weights less than 2500 grams. 

• We use only birth weight and we don’t have other information to diagnose FGR as a result we don’t know who are growth restricted or constitutionally small

• 18 Mothers were induced for obstetric and medical complications.

2. If at least one ANC(not referred and booked) follow-ups, considered as having ANC follow up but those who came with a referral from other institutions(151) will be asked whether they have ANC follow-up or not. If they have then how many follow-ups.

3. Congenital anomalies were excluded from the study.

Response for the second reviewer

1. I humbly accept all General comments and thank you for your constructive comment and then they all are corrected 

Reference number 27…mentioned at the sample size estimation part 

2. Abstract: Comments are Accepted and corrected 

3. Introduction: Classification of the cesarean section depends on different factors like Emergent or non-emergent cesarean sections and is based on women’s request in some countries which is not accepted in our country unless the Mather had cesarean section scar. The other classification is the Robson classification of a cesarean section which is recommended by WHO.

• The reference for introduction paragraph 3 line one is reference number is 11

4. Methods: comments are accepted

• Negative fetal heartbeat: intrauterine fetal death cases at admission were excluded

• Those babies, who were healthy, were usually discharged after 24 hours. Since the study includes the first 7 days, we just contacted the mother on the phone whether the baby is breastfeeding well, alert, sleeping well, having fast breathing, and fever. We do this only for healthy babies but if there is any problem, the mother was informed to bring it back.

• The Mather was informed about the study, that there will be follow up upon discharge till 7 days, while she is in the hospital

• Discharged neonates were followed by data collectors assigned to specific neonates.

5. Results: comments are accepted

• Cause of neonatal death were taken from neonatal intensive care unit registration book and we don’t have set up to autopsies so we commonly reach the conclusion by clinical characteristics

6. Discussion: comments are accepted and corrected

7. Consent: comments are accepted and corrected

8. References: comments accepted and corrected

---

## [Editor Report · Decision Letter 1]

28 Jan 2022

DETERMINANNTS OF EARLY NEONATAL OUTCOMES AFTER EMERGENCY CESSARIAN DELIVERY AT HAWASSA UNIVERSITY COMPRHENSSIVE SPECIALISED HOSPITAL, HAWASSA, ETHIOPIA

PONE-D-21-28401R1

Dear Dr. Tantu

We’re pleased to inform you that your manuscript has been judged scientifically suitable for publication and will be formally accepted for publication once it meets all outstanding technical requirements.

Kind regards,

Francesca Crovetto

Academic Editor

PLOS ONE

---

## [Editor Report · Acceptance letter]

15 Mar 2022

PONE-D-21-28401R1 

DETERMINANTS OF EARLY NEONATAL OUTCOMES AFTER EMERGENCY CESAREAN DELIVERY AT HAWASSA UNIVERSITY COMPREHENSIVE SPECIALISED HOSPITAL, HAWASSA, ETHIOPIA 

Dear Dr. Tantu:

I'm pleased to inform you that your manuscript has been deemed suitable for publication in PLOS ONE. Congratulations! Your manuscript is now with our production department. 

Kind regards, 

on behalf of

Dr. Francesca Crovetto 

Academic Editor

PLOS ONE